# Related adjustment of firm production after demand shocks

**Károly Miklós Kiss**[1,2☉]*, **László Lőrincz**[1,3,4☉], **Zsolt Csáfordi**[5‡], **Balázs Lengyel**[1,3,4‡]

**1** Centre for Economic and Regional Studies, Eötvös Loránd Research Network, Budapest, Hungary, **2** University of Pannonia, Veszprém, Hungary, **3** Corvinus University of Budapest, CIAS NeTI Lab, Budapest, Hungary, **4** Agglomeration and Social Networks Lendület Research Group, Eötvös Loránd Research Network, Budapest, Hungary, **5** Erasmus University Rotterdam, Rotterdam, Netherlands

☉ These authors contributed equally to this work.
‡ ZC and BL also contributed equally to this work.
* kiss.karoly.miklos@krtk.hu

## Abstract

Multiproduct firms often diversify into technologically related activities to exploit efficiencies of joint production; however, unrelated products in the company's portfolio provide access to distinct markets and can help to avoid industry-specific shocks. Yet, the underlying mechanisms of related and unrelated diversification are still poorly understood. Here we investigate diversification decisions of firms in periods when corporations' markets are hit by a demand shocks. In these times, cost efficiency considerations might drive firms to reduce costs by narrowing product portfolios and focusing on combinations of technologically related products, in which economies of scope and mutual capabilities can be exploited. To test this hypothesis, we consider two measures of demand shocks, decreasing sales volumes on the product market and increasing import competition; and analyze their association with changes of product portfolios of Hungarian firms in the 2003-2012 period. We find that production has become more coherent in terms of technological relatedness after firms were exposed to demand shocks. Evidence suggests related adjustment of firm production after demand shocks such that products unrelated to firms' core product are dropped from the portfolio but related products are added.

## 1. Introduction

Multiproduct firms are thought to exploit economies of scope by diversifying into technologically similar activities because they are motivated to introduce new products that require relatively few extra investments [1–3]. Technological similarities, also termed technological relatedness [4], are therefore of central concern in strategic decision of firms and in a wide variety of distinct economics literature including firm expansion, international development, and regional diversification [1, 5–8]. A growing body of empirical evidence suggests that related diversification of firms is the rule of expanding production portfolio [4, 7–13]. Further, the same logic has been applied to explain the development of industries [14–16], countries [17, 18] and regions [19, 20] by claiming that these units diversify easier to technologically related activities than to unrelated ones (for a recent review see [21]).

**Data Availability Statement:** The data that support the findings of this study are available from Central Statistical Office of Hungary (HCSO). Restrictions apply to the availability of these data, which were used under license for this study. HCSO opens up data files for those requesting data after an

accreditation process. Data is available upon request. We declare that the authors received no special access privileges and researchers would have access to the data in the same method as the authors. Upon request, the data bank of the research institute (Centre for Economic and Regional Studies) can provide access to the data. Contact: adatbank@krtk.hu or kollo.janos@krtk.hu.

**Funding:** The study was supported by a grant from National Research, Development and Innovation Office (grant no. K 135195). The funders had no role in study design, data collection and analysis, decision to publish, or preparation of the manuscript.

**Competing interests:** The authors have declared that no competing interests exist.

The synergies and complementarities across production classified in different industries is the major underlying argument behind the principle of relatedness. Several studies claim that firms can exploit these synergies by creating an appropriate product portfolio [2, 22–25], because product portfolios that consist of technologically similar products create efficiency gains for the firm.

Nevertheless, there are also many multi-product firms that produce technologically non-similar products. Nathanson and Cassano [26], for example, have found nearly the same number of firms with related and unrelated product portfolios. A key motivation for such portfolio may be that unrelated diversification is associated with resilience against industry-specific shocks both for firms [27] and for regions [28].

Our aim in this paper is to better understand the dynamics of related and unrelated diversification of firms. The question is, under what conditions do firms diversify into related activities and increase the coherence of their product portfolio, and when do they diversify into unrelated products? Although the question about the dynamics of diversification was raised long ago [29], we have few systematic empirical evidence on the factors that facilitate related versus unrelated diversification of firms' production.

We propose that investigating diversification decisions of firms when corporations' markets are hit by demand shocks versus their prosperity era offers new insights into the underlying mechanisms. When experiencing negative shocks, decreasing resources and cost efficiency considerations might press managers to reduce costs by narrowing product portfolios [30, 31] and focusing on combinations of technologically similar products, in which economies of scope and mutual capabilities can be exploited. In the absence of such forces, however, they might have more freedom to diversify into unrelated products, as a hedging strategy against market fluctuations [27] or as a strategy to secure their own employment positions [32]. Thus, the paper contributes to the literature by providing new evidence on how technological relatedness influences firms' decisions on product portfolio under different economic conditions.

The empirical case of the paper is provided by the universe of Hungarian firms. Our data enables us to observe production portfolio of every firm in industry sectors during the 2003-2012 period. To measure negative demand shocks, we apply two strategies. First, we utilize the heterogeneity of product market trends and identify product-level demand shocks from the product-level volatility in sales. Second, we estimate import substitution by Chinese import penetration in the EU, also on the product level.

The investigated period covers interesting heterogeneity of demand shocks across product markets and firms. Trade liberalization opened the Hungarian industrial market in the 1990s and has concluded after the country joined the EU in 2004. This opening hit some of the traditional industries. For example, a significant decline was observed in the food industry due to the adverse effect of EU agricultural agreements, and in the and textile industry due to a reallocation mechanism that destroyed less productive firms [33, 34] and to a significant increase of the minimal wage [35]. The period after EU accession is associated with increasing foreign direct investment, gross value added, and export orientation, which is interrupted in 2008-2010 by the global financial crisis that also hit industries to a different extent. After the financial crisis, a re-industrialization can be observed that is selective across industries. Transport equipment sector grew both in terms of employment and value-added, however, computers and electronics have decreased [36, 37].

During this period also an increasing competition with import from China was observed in general, with various extent across the different products. Accordingly, we use increased import competition as a second measure of demand shocks, which we also identify from its product-level variation.

Our empirical exercise reveals that production becomes more coherent in terms of technological relatedness when firms were exposed to demand shocks. However, the channels are somewhat different for the two types of shocks examined. We find that if import competition intensifies, firms *drop* peripheral products not related to their core product. These results are consistent with the predictions of models of heterogeneous multi-product trading firms [38, 39] suggesting that firms drop marginal products and focus on related product lines to increase their productivity as a consequence of increased import competition.

We also find that during decreasing markets firms diversify with high likelihood into those new products that are related to their portfolio. This is consistent with the assumption that decreasing markets force firms to better exploit of economies of scope by limiting them adding unrelated products for risk-sharing concerns or managerial motives of firm expansion. However, the results suggest, that once present, products are not removed from the portfolio, unless the firm operates on decreasing markets *and* is at the same time faces high levels of financial exposure.

The structure of the paper is as follows. In the following section, we summarize the theoretical background of product diversification and firms' strategies and propose research questions and hypotheses. Then, in Section 3, we introduce the structure of the data and present the methodology of the measurements and the empirical model. The results are discussed in Section 4 and main conclusions are drawn in Section 5.

## 2. Theoretical background and research questions

A common classification of product diversification distinguishes between related diversification, when the firm expands into new products that are technologically very similar to products already in the portfolio, and unrelated diversification, when similarities between existing and new products are low or do not exist. In this Section, we provide an overview of drivers of related and unrelated diversification, discuss the literature on firm strategies when facing negative shocks, and finally, present our hypotheses.

### 2.1. Drivers of related diversification

Related diversification is basically explained by efficiency motivations. Synergies between production processes may increase efficiency and motivate firms to diversify into related businesses [40–42]. Several studies suggest that related diversification is superior to unrelated diversification when the primary goal is to increase the efficiency, because synergies increase firm performance [43–46].

This argumentation highlights that multi-product firms make more efficient use of resources [47]. The underlying idea appears already in Penrose's [1] explanation for corporate growth: if firms have indivisible resources, which cannot be fully exploited in their original core business (excess capacity arises) and these resources can be utilized in other industries, it encourages companies to grow through diversification. The theoretical foundation of the underlying cost-efficiency arguments is the concept of economies of scope [22, 48–50]. According to Panzar and Willig [50], the source of economies of scope is the sharing or joint utilization of inputs. It arises if a given input is imperfectly divisible and its utilization in producing a single product would leave excess capacity. Further, certain resources, such as information, skills or knowledge do not deteriorate; therefore, they are also available freely for other production processes. Indivisible (or not fully divisible) resources incur fixed costs (or quasi-fixed costs) [22, 50, 51]. If these fixed costs are divided across various products in case of multi-product firms, then they result in lower unit costs.

Teece [2] emphasizes that the resource-based explanation of diversification always assumes the presence of transaction costs. If the product and capital markets worked perfectly, these efficiency aspects could not justify the appearance of multi-product firms. If there were no transaction costs, excess capacity of indivisible resources could be efficiently sold or leased in the market, thus the cost advantages behind the economies of scale and scope could be exploited in the same way for single-product firms.

Another economic mechanism explaining the efficiency of joint production of products within firms is complementarity. Weiss [52] emphasizes that the common notion of equating relatedness with similarity is incomplete because it does not take resource complementarity into account when the joint use of distinct resources produces a higher total return than the sum of returns of independent utilization of each resource. Consequently, firms primarily search for complementary resources rather than similar resources during acquisitions, because combination and integration of the acquirer's resources with target firm's resources might increase the efficiency of resource utilization [53, 54].

## 2.2. Drivers of unrelated diversification

Explanations of unrelated diversification can be classified into three major groups: risk reduction motives, motivations to increase market power, and individual managerial motives.

Several studies [55–58] argue that in case capital markets are not efficient, there may be financial motives for unrelated firm expansion, that is, the product diversification can be an *element of risk reducing strategies*. Firms can allocate their activities across industries with opposite market risks to reduce the exposure of their core business. Amit and Livnat [59] revealed that benefits of unrelated diversification include more stable cash flows, lower operating risks and increase the levels of leverage. Galbraith et al. [27] showed that in R&D-intensive industries unrelated diversification is frequently applied as a hedging strategy against unforeseen technological shocks. Like the economies of scope theory, this explanation also requires market failures and transaction costs in capital market, otherwise these risks can be reduced by allocating investments among various financial instruments. Indeed, Dundas and Richardson [29] revealed that market failures in product and technological markets motivate the firms to diversify into related industries, whereas imperfections in capital markets raise the probability of unrelated diversification. Robson et al. [60] observe a similar risk-sharing motive of diversification in small firms. They argue that diversification is often used by small firms as a risk-spreading survival strategy rather than a growth strategy. The authors find a negative relationship between diversification and growth in the case of small firms. They conclude that small entrepreneurs do not have the necessary resources and skills to manage diversified activities and therefore diversification can be considered a risk-sharing strategy in small firms, these entrepreneurs tend to diversify to compensate a decline in their original core business.

Other explanation of unrelated diversification is firms' strategy towards increasing *market power*. The literature of conglomerates emphasizes the anti-competitive effects of the formation of large multi-industry companies operating in unrelated industries [61–64]. According to these explanations, operating in unrelated industries allows companies to engage in anti-competitive practices that include cross-subsidization, when the profit earned in one market allows maintaining predatory pricing in another market [57], and mutual forbearance, when big conglomerates compete in multiple markets and recognizing this interdependence, they mutually respect each other's dominant markets, competing less vigorously [64].

Finally, explanations of other *individual managerial drivers* are based on the agency problem arising from the separation of ownership and management. Shleifer and Vishny [65] present that managers use strong corporate expansion as a tool to increase the firm's demand for

their personal skills. Amihud and Lev [66] argue that managers also use corporate expansion (unrelated conglomerate mergers) to reduce their employment risk. While corporate owners can reduce their risks by creating sufficient capital market portfolios, corporate executives can only reduce their employment risk by expanding the company's activities. Hence managers use diversified expansion as a means of reducing total firm risk and at the same time their employment risk and smoothing the earnings stream, which could lead to excessive diversification. Accordingly, they find that manager-controlled firms are more diversified than owner-controlled firms and that the conglomerate type (i.e. unrelated) expansion is more frequent in case of manager-controlled firms than owner-controlled firms. Denis et al. [32] show that managerial ownership is negatively related to diversification, and that market disciplinary forces, such as managerial turnover or financial distress increase corporate focus. Gomez-Mejia et al. [67] finds that family firms diversify less than non-family firms.

### 2.3. Firms' strategies when facing negative shocks

Firms can choose different strategies when facing difficult economic conditions due to environmental shocks, economic recessions, and declining industries. Kitching et al. [30] distinguish an investment-based expanding strategy, as against the cost reducing adaptation to crisis. As they show, some firms perceive crisis periods as opportunities to invest, innovate and expand into new markets. However, most firms are not able to implement such investment-based expanding strategy, because it requires financial and managerial resources. Firms hit harder have limited resources and focus more on short-term survival, choosing cost-cutting strategies more likely [30] and in fact, the most common behavioral response of firms to falling demand is cost cutting [68]. These retrenchment strategies can consist of a wide range of cost reducing actions: divestment of assets, establishment closure, labor cost reduction (downsizing or working hours reductions), cutting expenditures of non-core activities such as R&D, marketing, and employee training [30]. This strategy is the typical for small firms [69], and it also seems to improve their performance [70].

Despite cost reduction is the most frequent strategy of firms when facing negative demand shocks (e.g. in times of economic downturns), some authors debate its' effectiveness. They argue that those firms that cut costs in order to survive in the short-run must face long-run opportunity costs of capacity reduction, because they might be unable to adapt adequately during economic recovery [71]. In addition, when demand is lower, opportunity costs of investing in productivity-enhancing innovations may be lower [72], and in crisis times payoffs from investments like R&D and marketing may be higher, as there is lower level of such activity on the market due to budget cuts [73].

Increasing import competition can be another cause of industry-level demand shocks. Imported products are substitutes for domestically produced products, causing firms to face declining demand for their products and stronger price competition. Several studies examined the impact of increased exposure to imports from low-wage countries on labor market outcomes. It was found that increasing imports from China is negatively associated with the level of employment, labor force participation, and wages in import-competing manufacturing industries [74–77]. Bernard et al. [78] find that survival and growth rate of those US manufacturing plants decreased which has been exposed to increasing industry-level import competition from low-wage countries. Bloom et al. [79] show that increasing Chinese import competition has led to a decline in employment, profits and prices. Dobson and Waterson [38] show that intensified competition may encourage firms to specialize and thus limit their product range. There are also studies however that highlight the positive effects of Chinese imports on European firm performance. Such effects may arise across different product groups

within the same firm through providing cheaper but high-quality intermediate inputs thus facilitating a more efficient production [80–82].

## 2.4. Research questions and hypotheses

Previous studies suggest that firms frequently choose cost reducing retrenchment strategies as a response to the falling demand and expected profit, decreasing cash flow and lack of financial resources [30, 68–70]. Accordingly, we can suppose that facing demand shocks makes the cost efficiency concerns more important for firms, which will limit the options of managers. In the context of product diversification, efficiency-based drivers – the needs for more efficient use of resources – encourage utilization of economies of scope which leads to related diversification, while in the case of non-efficiency driven corporate expansion (driven by risk reducing motives or individual managerial and market-power-based drivers), technological proximity of industries is either irrelevant or even encourages unrelated diversification.

We can thus expect that firms rather choose the exploitation of economies of scope, while other drivers of diversification (risk-sharing concerns or other managerial motives of firm expansion) are pushed into the background when they are exposed to negative demand shocks. To formulate a hypothesis around these claims, we measure the technological coherence of product portfolios by tools of technological relatedness such that a product portfolio is coherence in case it contains technologically related products. Negative demand shocks are measured by decreasing markets and increased import competition. These measurements are described in the next section and captures the potentials of economies of scope.

*H1a*: *When exposed to negative demand shocks, the technological coherence of company portfolios increase.*

However, opportunities of managers to choose between different strategies may also be influenced by endowments of the firms. Therefore, we expect that firms that have less tight liquidity constraints are less pressed towards exploiting the short-term efficiency benefits of the related product portfolios. Therefore:

*H1b*: *Demand shocks increase technological coherence of product portfolios more intensively in those firms that face liquidity constraints.*

When negative shocks hit in, firms' strategies can include retrenchment and expansion as well. Thus, one can investigate the dynamics of product portfolios and distinguish events of dropping and adding products. Based on the previous argument, we expect that efficiency arguments will gain more importance in both types of crisis situations compared to prosperous times. We can formulate our hypotheses accordingly:

*H2. When exposed to negative demand shocks, firms are more likely to drop products that are technologically less related to their core production than more related products.*

*H3. When exposed to negative demand shocks, firms are more likely to add more products related to their core production than unrelated products.*

## 3. Methods and data

### 3.1. Data

We use the Hungarian PRODCOM database for years 2003-2012. PRODCOM data is based on reporting the production volumes and sales values for each product the firm produces to the Statistical Office on a yearly basis. This reporting covers (1) all companies that have

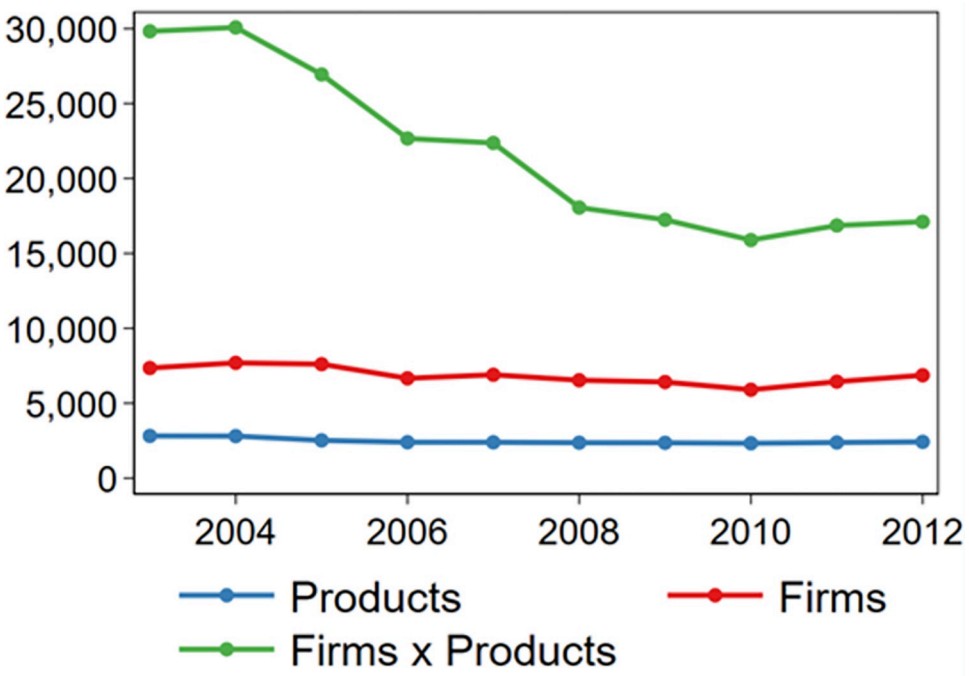

**Fig 1. Number of firms and products in the analysis.**

more than 20 employees operating in industrial sectors, (2) a sample of industrial companies employing 6-19 workers, and (3) all firms having more than HUF 500 million (€ 1.65 million) sales revenue from industrial products, even if they are not classified to the industrial sector by their primary activity. This results in a sample of approximately 7,000 firms and 2,400 products (Fig 1).

This sample in terms of number of firms seems to be relatively small compared to the number of all firms operating in Hungary, that were around 550,000 in the observed period (EUROSTAT). Approximately 90% of Hungarian firms are however very small, having less than 5 employees (STADAT, Hungarian Statistical Office [83]; therefore, they are not included in the PRODCOM database. Comparison of the PRODCOM data to all enterprises with more than 5 employees (Fig 2) reveals that it indeed covers more than 90% of the medium and large firms operating in the manufacturing and in the mining sectors, and more than 50% of the firms in the 10-50 employee category. Distribution by industry shows that 95% of the observations come from manufacturing industries. Despite firms from the mining and from the energy and water sectors also have high probability of being included, they represent a small minority (3%) of the data, due to the relatively low number of such firms in Hungary compared to manufacturing. Firms registered in other sectors (agriculture, construction, trade and services) constitute only 2% of the observations.

Although the typical firms in the data employed 11-50 workers, due to the presence of large firms, the average firm size was 102 for the firms in our data. Firms created 72.9 thousand USD value added per year per worker on average. They operated with a 3% profitability ratio (compared to their total revenue), and on average 38% of their equity was coming from short-term debts (Table 1). Firms are relatively young, operating for 11.5 years on average, because 98% of them were established after the transition in 1989.

Data to the import competition variable were accessed from the following sources:

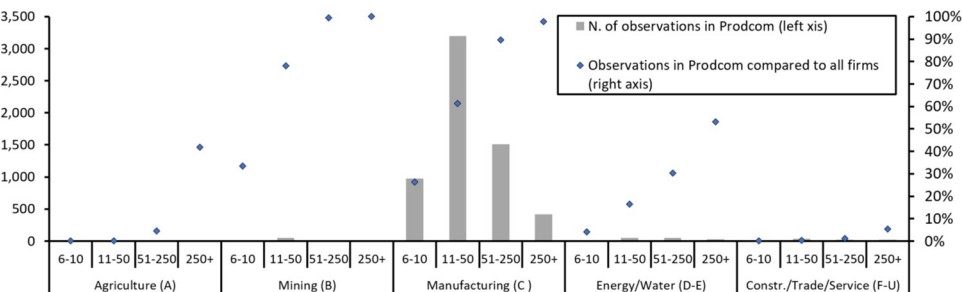

**Fig 2. Number of observations according to firm size and industry with comparison to all firms.** Note: Observations represent mean values for 2003-2012. Source: Authors' calculations based on the PRODCOM database and the balance sheet data provided by the Hungarian Statistical Office.

1. Trade value of imports from China to EU28 countries on the HS2002 6-digit product level for years 2002-2006, and on the HS2007 6-digit product level for years 2007-2011 from Comtrade.un.org.

2. EU28 Consumption was proxied as Sold production (Sales) + Import – Export, for which data was accessed from Eurostat's Prodcom database.

## 3.2. The measurement of technological relatedness

To measure technological relatedness between *products*, we apply the revealed relatedness methodology developed by Neffke and Henning [84]. This method calculates relatedness measures between *industries* and assumes that they are technologically related in case firms often produce the combination of products that belong to these industries.

Relatedness between products is calculated on the industry-level, which is equivalent to the first 4 digits of Prodcom product codes that define the 4-digit NACE industry codes. Therefore, industries can be viewed as product groups. This level of aggregation makes the results comparable with previous studies using this method [20, 84, 85]. Calculating relatedness across 6-digit Prodcom codes would introduce too much noise to the measurement and the indicators would turn highly volatile when decomposing it to years. Accordingly, in the first step of our data procession, we assign the *products* of the firms to *industries*.

Next, the core product of each company is identified. We define this as the product that the biggest part of the firm's total revenue stems from. We applied this definition on yearly basis; therefore, it is possible that a firm changes its core product over time. Formalizing the definition, let $Y_{ipt}$ stand for the sales value of firm $i$ from product $p$ in year $t$, then for firm $i$ across its products $p$, the firm's core product in year $t$ is $c_{it} = p$, where $Y_{ipt} = \max_{it}(Y_{ipt})$.

**Table 1. Key characteristics of the firms.**

| Measure | Mean | S.d. | N |
|---|---|---|---|
| *Size of employment* | *102.8* | *428.2* | *64,877* |
| Firm age | *11.46* | *5.45* | *63,524* |
| Productivity[1] | *72.9* | *284.5* | *64,840* |
| *Profitability* | *0.030* | *0.133* | *63,217* |
| *Short-term debt ratio* | *0.376* | *0.222* | *60,787* |

Notes: 1. Yearly value added by employee in 1000 USD.

Third, we define the industry of the firms: each company is assigned to a 4 digit industry code, which corresponded to its core product in the PRODCOM classification.

This classification enables us to calculate technological relatedness of industries *A* and *B* as the frequency that firms produce two or more products such that at least one belongs to industry *A* and at least another one belongs to industry *B*. A directed link from industry *A* to industry *B* is created, if a company's core product belongs to sector *A*, but it also produces products, which belong to sector *B* according to the PRODCOM classification. By summing the links over the industries, we got the co-occurrence network of industries, where each node corresponds to a 4-digit industry, and each edge weight corresponds to the number of companies active in production in both industries.

As we discussed above, diversification into products can be motivated not only by economies of scope and technological proximity. Other factors, such as high expected profits or industry size are important as well so that firms might decide to expand their activities to attractive industries. To control for other factors that influence the number of co-occurrence links we follow Neffke and Henning [84] and estimate the expected number of co-occurrences for each pair of industries using industry specific characteristics as predictors (e.g. profitability, sales volume, competition expectations, wage level etc.). Then the difference between the predicted and observed co-occurrences will reveal the relatedness between each industry pair concerned. As most of the potential pairs of industries do not occur along with each other in the same firm, i.e. there is an extensive number of zeros, we follow Neffke and Henning [84] who suggest a zero-inflated negative binomial regression to estimate the level of co-occurrence. The model is formulated as follows:

$$E(L_{ij}|v_i, w_j, \varepsilon_{ij}) = [1 - \Pi_0(\gamma + \delta_i v_i + \delta_j w_j)]e^{\alpha + \beta_i v_i + \beta_j w_j + \varepsilon_{ij}}, \tag{1}$$

where $L_{ij}$ is the number of actual co-occurrence links from industry *i* to *j* and $v_i$ and $w_j$ are the specific characteristics (number of active firms, average profit level, income level, firm size in employees, and value added) of industry *i* and *j*.

From the predicted values of the regression, we could get our revealed relatedness indicator:

$$\hat{RR}_{ij} = \frac{L_{ij}^{obs}}{k\hat{L}_{ij}}, \tag{2}$$

where $\wedge$ indicates fitted value and *k* is a normalizing constant.

The matrix of *RR* indices can be visualized as technological proximity network of industries that we use to explain firms' diversification behavior and the dynamics of the product portfolio during crisis (Fig 3).

One might raise the concern that such a revealed technological relatedness measure might be distorted by the presence of: 1) conglomerate firms that choose to differentiate their production lines in structurally unrelated markets; 2) vertically integrated firms that choose to produce the inputs necessary for their production. Presence of conglomerated and vertically-integrated firms would not be a problem if we had site-level data, as firms produce technologically related products at the same site, which can benefit from the resulting cost efficiency in production. Unfortunately, our database only contains firm-level data. However, two things reduce the problem raised. First, to avoid this multi-plant problem, our analysis focuses on the manufacturing sector. Békés and Harasztosi [86] show that in the Hungarian manufacturing industries over 90% of firms have one site only, which suggests that the bias is not significant. Moreover, as shown above, the difference between the predicted and observed co-occurrences reveals the relatedness between each industry pair. The relatedness indicator between two industries would be significantly distorted if both industries were composed of very similar

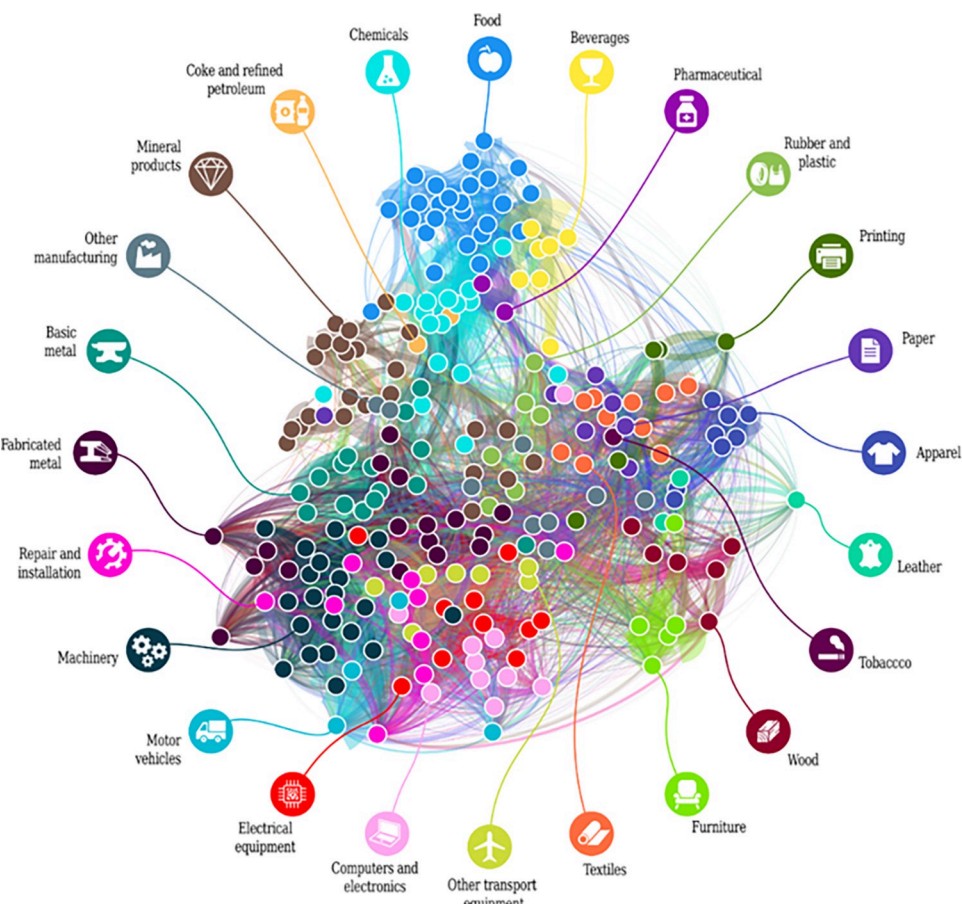

**Fig 3. The technological proximity network of manufacturing sectors.** Nodes depict manufacturing sectors according to NACE (Rev. 2) 4-digit level codes, edges represent the average revealed relatedness value for the period 2003-2012. Color codes have been set according to NACE (Rev. 2) 2-digit industries. Position of the nodes is determined by the Force Atlas 2 algorithm of Gephi [87].

conglomerates or vertically integrated companies expanding in the same unrelated directions. Thus, the problem is considered legitimate, but the arguments above suggest that the bias does not really give cause for concern.

The technological proximity network of Hungary (Fig 3) reveals that those 4-digit industries tend to cluster that belong to the same 2-digit NACE industries (we made available the technological proximity data at: http://www.mtakti.hu/relatedness/). For example, majority of subsectors in the food industry cluster together in the upper part of the network. Additionally, one might also find logic behind the relative position of industries in the network in two terms. First, there is a relatively dense core of the network including machinery, basic metals etc; while there are loosely connected peripheries in the network including food, apparel, textiles, wood etc. industries. Second, those industries that one assumes to be similar by intuition (e.g. wood and furniture) are indeed neighbors.

### 3.3. Measuring diversification

The technological coherence of firms' production portfolios is the main object of the first hypothesis. For the measurement, we identify the core product of the firm as the product that

the biggest part of the firm's total revenue stems from. The coherence of firms' product portfolios is defined as the arithmetic mean of relatedness between the firm's core product and all additional products:

$$Coherence_{it} = \frac{\sum_{p=1}^{m} RR_{cp,t-1}}{m}, \forall p \neq c \tag{3}$$

where $c$ is the core product of firm $i$ (for simplicity from here on we omit the $it$ indices from the notification of the core product and only refer to it a $c$ instead of $c_{it}$.), $m$ is the number of products the firm produces, $p \neq c$ represents its non-core products, $Y_{ipt}$ is the sales value of firm $i$ from each product $p$, and $RR_{cp,t-1}$ is the relatedness index between the industries of product $p$ and $c$ in the previous year.

We apply the lag of relatedness that is necessary to avoid potential endogeneity. Since we calculate relatedness from average product portfolio of firms, the measure without time-lag would be problematic to use to explain the decision of firms on product portfolio. The time lag allows us to examine how the average relatedness index of the product portfolio measured in the previous year changes in the current year, for example, in response to a demand shock.

## 3.4. Measurement of negative demand shocks

Our analysis focuses on the relationship between product-level demand-shocks and production portfolios, and thus we identified our measures on the product level. At the first instance, we define product-level *Market shrinkage* based on production volumes. Secondly, we define product-level *Import shock* based on increasing share of Chinese imports.

For simplicity, we measure the market shrinkage using the relative change of its production value from the previous year over all the producing firms. Thus, our product-level measure is:

$$Market\ shrinkage_{pt} = -\frac{\Delta Y_{pt}}{Y_{p,t-1}} \tag{4}$$

where $Y_{pt}$ is the sales value of all firms producing product $p$ in year $t$. Note that more severe market shrinkage is indicated by positive values, while negative values indicate expansions of the market or prosperity. To measure a firm's exposure to decreasing markets, we calculate firm-level (weighted) averages of the product-level crisis over the $m$ products, a firm produces:

$$Shrinkage\ exposure_{it} = \frac{\sum_{p=1}^{m} Market\ shrinkage_{pt} Y_{ipt}}{\sum_{p=1}^{m} Y_{ipt}}. \tag{5}$$

Note that we measure market shrinkage (or prosperity in case of negative values) based on production value of the Hungarian market, which also includes the observed firm. This creates mechanical correlation between our dependent and independent variables. To avoid potential endogeneity, we will use the lagged version of our crisis measure to predict diversification in the following year. However, if the firm's decisions are correlated over time, or unobserved factors' influence them, this still can be problematic. We have no such concerns about our other measure the exposure to import competition from China.

To measure import shocks, we follow the approach of Bloom, Draca and van Reenen [79] to proxy the shocks by the increasing export competition from China. We utilize that Hungary is a relatively open economy; over 60% of the produced products are exported, and it is very strongly connected to the European Union; over 75% of its exports are going to the other EU countries. Therefore, the increase of Chinese imports in the EU may have significant adverse

effects on the Hungarian producers. Meanwhile, Hungary is a very small player compared to China even on the EU markets. The EU imported products of 375,000 Million USD from China in 2010, while the whole Hungarian export was about a quarter per cent of this value (94 Million USD). Therefore, reverse causality is not likely; the behavior of Hungarian firms hardly influences prices and production volumes in China.

We define our product-level import shock measure as the increase in Chinese imports in the products at the EU markets. The percentage change of imports from China to the EU28 countries from the previous year's value was calculated on the Prodcom 6-digit level and standardized by the EU28 consumption to correct for effects that may come from the general decline of EU markets in the respective product group:

$$Import\ shock_{pt} = \frac{\Delta IMP^{CH}_{pt}}{IMP^{CH}_{p,t-1}} \tag{6}$$

where $\Delta IMP^{CH}_{pt}$ denotes the percentage change in the ratio of Chinese imports to total consumption in the EU for product $p$ in year $t$. Note, that by this measurement of import shocks based on product-level imports we also avoid mixing up the increased product competition effect of Chinese imports with their positive effect of making the intermediate goods cheaper, which we should have been addressed if choosing other import shock measure, such as tariff levels.

For our analysis, we calculate the firm-level exposure to increasing import shocks from China by the weighted averages of the import shocks of the different products in the firm's portfolio:

$$ImpShock\ exposure_{it} = \frac{\sum_{p=1}^{m} Import\ shock_{pt} Y_{ipt}}{\sum_{p=1}^{m} Y_{ipt}} \tag{7}$$

where $Y_{ipt}$ denotes the sales of firm $i$ of product $p$ in year $t$, whereas $m$ denotes the number of products the firm produces. Thus, $Y_{ipt}$ is used as weights in calculating the weighted average.

## 3.5. Statistical methods

We apply three methods to analyze the relationship between demand shocks and related diversification. In the first exercise, we look at how coherence of the product portfolio changes in firms according to their exposure to negative demand shocks measured by shrinking markets and import shocks (*Hypothesis 1a*). In this model, we also examine the impact of liquidity mechanism: whether a restricted liquidity amplifies the impact of the demand shock on the coherence of the product portfolio for firms more exposed to financial markets (firms with higher short-term debt) (*Hypothesis 1b*). Next, we analyze the events of dropping products from the portfolio, and check, whether relatedness is a stronger predictor of keeping a product conditionally to the exposure to negative demand shocks (*Hypothesis 2*). At last, we analyze the events of adding new products under identical conditions (*Hypothesis 3*).

In the first approach, we use firm-level panel regressions with firm fixed effects. The firm fixed effects enable us to exclude alternative explanations based on unobserved heterogeneity of firms (e.g. those industries that exhibit more related diversification were also hit harder by shrinking markets or import shocks), as we compare the coherence of the product portfolio in the firm in the year following the identified demand shock with its previous values. Our explanatory variables are the exposure to shrinking markets, or to import shocks. We also add time dummies ($D_t$) to control for yearly fluctuations in the trends. To assess the impact of financial exposure of the firms, we also add the *Leverage* ratio (short term debt by equity) to

the model, together with its interaction with our main independent variables, *Shrinkage exposure* and *ImpShock exposure* to the market shrinkage model. We estimate:

$$Coherence_{it} = \beta_0 + \beta_1 Coherence_{i,t-1} + \beta_2 Shrinkage\ exposure_{i,t-1} + \beta_3 Lever_{i,t-1} + \beta_4 Lever_{i,t-1}$$
$$\cdot Shrinkage\ exposure_{i,t-1} + \beta_5 D_t + \xi_i + \varepsilon_{it} \tag{8}$$

and

$$Coherence_{it} = \beta_0 + \beta_1 Coherence_{i,t-1} + \beta_2 ImpShock\ exposure_{it} + \beta_3 D_t + \xi_i + \varepsilon_{it} \tag{9}$$

where $D_t$ are year dummies, and $\xi_i$ and $\varepsilon_{it}$ are error terms.

The second hypothesis is tested by examining the likelihood of dropping products conditional to their relatedness to the company's core product. Our dependent variable is the dummy of dropping a non-core ($p \neq c$) product from the portfolio of a firm ($i$) in year $t$, $Drop_{i, p \neq c, t}$. The key independent variable is the value of relatedness $RR_{cp,t-1}$ of the examined product to the core product of the firm, one year prior to the decision. Our specific interest is whether the effect of relatedness is stronger when demand shocks are experienced than in other instances; thus, we include the interaction of the relatedness value with the Market shrinkage and Import shock terms. As controls, we add the main effect of the Market shrinkage and Import shocks, the Leverage ratio and its interactions to the Market shrinkage variable, and year dummies ($D_t$) to both models:

$$Drop_{ipt} = \beta_0 + \beta_1 RR_{cp,t-1} + \beta_2 Market\ shrinkage_{pt} + \beta_3 RR_{cp,t-1} \cdot Market\ shrinkage_{pt} + \beta_4 Lever_{i,t-1}$$
$$+ \beta_5 Lever_{i,t-1} \cdot Market\ shrinkage_{i,t-1} + \beta_6 RR_{cp,t-1} \cdot Lever_{i,t-1} \cdot Market\ shrinkage_{i,t-1} + \beta_7 D_t + \xi_{ic}$$
$$+ \varepsilon_{ipt},\ \forall p \neq c \tag{10}$$

And for product-level import shocks:

$$Drop_{ipt} = \beta_0 + \beta_1 RR_{cp,t-1} + \beta_2 Import\ shock_{pt} + \beta_3 RR_{cp,t-1} \times Import\ shock_{pt} + \beta_5 D_t + \xi_{ic}$$
$$+ \varepsilon_{ipt},\ \forall p \neq c \tag{11}$$

We use fixed-effect panel regressions with the fixed-effects on firm-core product levels to compare only those instances within firms when they did not change their core products. $\xi_{ic}$ is the corresponding error term. For simplicity, we used linear probability models to estimate Eqs 10 and 11. Note that the units of analysis are firm-products; therefore, our identification comes from comparing products that the firm produces in the given year and in other years.

The estimations defined in Eqs 10–11 may raise concerns of endogeneity. Namely, we predict firms' decisions on their product portfolio with the value of relatedness that we observe in the portfolio, which in turn comes from diversification decisions of firms, including the focal firm as well. Once again, we can address this issue by lagging relatedness with one year. Thus, the indicator will capture the tendency that the given products have been produced jointly in all firms of our sample with the core product of the firm that we investigate in the regression. This "tendency" certainly includes the previous activity of the examined firm as well. However, we compare products, which were *both* included in the product portfolio in the previous year. Thus, the difference between their relatedness to the core product must come from the activity of other firms on the market. In sum, we argue that lagging relatedness solves the direct endogeneity issue.

To test diversification into new products, we estimate the relation between relatedness and adding products in situations according to firms' exposure to demand shocks. The dependent variable is $Add_{ipt}$ that takes the value of 1 if the product was added to the portfolio of the firm

and zero otherwise:

$$\begin{cases} Add_{ipt} = 1 \ if \ (Y_{ipt} > 0 \ and \ Y_{ip,t-1} = 0) \\ Add_{ipt} = 0 \ if \ (Y_{ipt} = 0 \ and \ Y_{ip,t-1} = 0) \end{cases} \qquad (12)$$

Thus, each product, that were not part of the firm's product portfolio in the previous year, must be considered for each year and for each firm. For reasons of simplicity, we exclude those products that were not introduced by any firms in the examined year.

To avoid mechanical correlation, we modify the Shrinkage exposure and ImpShock exposure variables for this model by excluding the newly added product from the firm-level average, thus considering only those products that were already in the firm's portfolio:

$$Shrinkage \ exposure'_{it} = \frac{\sum_{p=1}^{m} Market \ Shrinkage_{pt} Y_{ipt}}{\sum_{p=1}^{m} Y_{ipt}}, \ \forall p | Add_{ipt} \neq 1 \qquad (13)$$

$$ImpShock \ exposure'_{it} = \frac{\sum_{p=1}^{m} ImpShock_{pt} Y_{ipt}}{\sum_{p=1}^{m} Y_{ipt}}, \forall p | Add_{ipt} \neq 1 \qquad (14)$$

Our key predictors are the lagged value of relatedness of each considered product to the core product ($RR_{cp,t-1}$), and these adjusted market shrinkage and import shock exposure variables. The hypothesis again considers the interaction terms of these variables. We add year dummies ($D_t$) to control for yearly fluctuations, and fixed effects for the core product of the firms ($\xi_c$). In addition, we add the leverage ratio and its interactions to the market shrinkage model. The estimated models are:

$$\begin{aligned} Add_{ip,t+1} = &\beta_0 + \beta_1 RR_{cp,t-1} + \beta_2 Shrinkage \ exposure'_{it} + \beta_3 RR_{cp,t-1} Shrinkage \ exposure'_{it} \\ &+ \beta_4 Lever_{it} + \beta_5 Lever_{it} \cdot Shrinkage \ exposure'_{it} + \beta_6 RR_{cp,t-1} \cdot Lever_{it} \\ &\cdot Shrinkage exposure'_{it} + \beta_7 D_t + \xi_c + \varepsilon_{ipt} \end{aligned} \qquad (15)$$

$$\begin{aligned} Add_{ip,t+1} = &\beta_0 + \beta_1 RR_{cp,t-1} + \beta_2 ImpShock \ exposure'_{it} + \beta_3 RR_{cp,t-1} ImpShock \ exposure'_{it} + \beta_4 D_t \\ &+ \xi_c + \varepsilon_{ipt}. \end{aligned} \qquad (16)$$

## 4. Results

### 4.1. Descriptive statistics

Descriptive statistics of our measures are displayed in Table 2. These distributions indicate that while the revealed relatedness between two randomly selected industries is rather low, coherence of actual firms' portfolios is much higher. The mean of the Market shrinkage measure is negative, indicating that firms on average enjoyed prosperity during the period, however, its high standard deviation signifies that many of them experienced negative demand shocks in different years. On the other hand, increasing Chinese imports was observed on average throughout the observed period, therefore the mean Import shock value is positive. It is also visible that the Market shrinkage measure is one magnitude lower than the Import shock measure, which is because Hungarian production data is more volatile than Chinese imports in the EU. Furthermore, we observe the coherence measure for fewer firms compared to Market shrinkage and Import shocks, because it can only be calculated for multi-product firms.

From the yearly averages illustrated in Fig 4 it is visible that the Market shrinkage follows a very different trend from the Import shock. Market shrinkage increased after Hungary's joining the European Union in 2004, then decreased in subsequent years. We see an increased

**Table 2. Descriptive statistics.**

| Measure | Mean | S.d. | N |
|---|---|---|---|
| Relatedness ($RR_{ij}$) | 0.0036 | 0.0351 | 60,939 |
| $Coherence_{it}$ | 0.171 | 0.207 | 32,913 |
| Market shrinkage$_{it}$ | -0.154 | 0.480 | 57,488 |
| Import shock$_{it}$ | 0.006 | 0.021 | 44,540 |

variance in 2008, then the mean value of the indicator jumped to its peak in 2009, and it was decreasing in the subsequent years. On the contrary, Chinese imports slowly increased in the EU, except in 2009, when we see a negative spike in the trend of the Import shock measure (Chinese exports are missing for 2012 because of translation ambiguity issues between SITC and CPA6 for this year). The Coherence indicator of firm's product portfolio fluctuates across the years in the period. As Import shock and Market shrinkage show somewhat opposite trends, and they also may impact different firms, we cannot draw clear conclusions from these aggregate descriptive findings on the correlation between the Coherence and these measures.

The decrease of Chinese exports to the EU countries during the financial crisis of 2008-2009 was primarily due to the decrease in GDP growth of the EU countries (most importantly its biggest markets in the EU: Germany, Netherlands, UK and Italy) [88]. Despite Hungary's economy is heavily linked to these countries, on the product level we do not observe significant correlations between the Import shock and Market shrinkage measures. We checked this in cross-section, in a pooled (OLS) model, and in a panel model with year FEs using different lags.

## 4.2. Firm coherence and crisis

Results indicate an increasing coherence of the portfolio after a firm was exposed to import shocks (Table 3), which supports Hypothesis 1a, but no such direct effect can be observed for being exposed to market shrinkage. The positive and significant interaction term of the Shrinkage exposure with the Leverage ratio confirms Hypothesis 1b. The insignificant coefficient of the Shrinkage exposure variable together with its positive and significant interaction term with the Leverage ratio suggests that when experiencing shrinking markets, those firms increase product portfolio coherence, which also have significant financial exposure (high Leverage). When ImpShock exposure is used as the main explanatory variable to explain next year's Coherence, we find that moving up 1 standard deviation in the shock variable is associated with an increase of 0.021×0.150=0.0032, i.e. 0.015 standard deviation in the Coherence of the firm's product portfolio in the next year (the coherence of previous year is only a technical variable in the model to serve as reference point for the next year and to capture the stability of

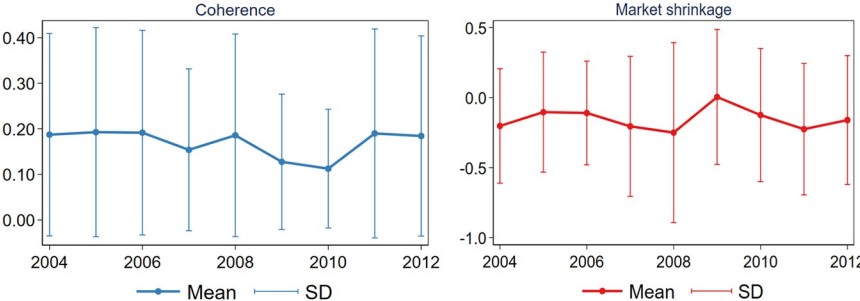
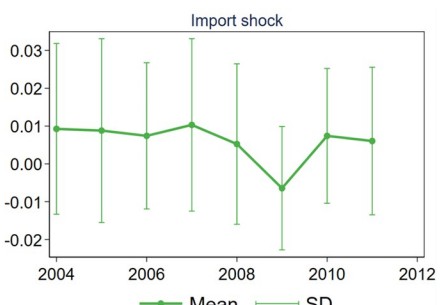

**Fig 4. Evolution of Coherence, Market shrinkage and Import shock measures.**

**Table 3. Change of the Coherence of firm's portfolios.**

| Dependent: | Coherence (t) | | |
|---|---|---|---|
| Shrinkage exposure (t-1) | 0.0002 (0.0032) | | |
| Shrinkage exposure (t-1) x Leverage (t-1) | 0.0157** (0.0074) | | |
| ImpShock exposure (t-1) | | 0.150*** (0.0364) | 0.163*** (0.0710) |
| ImpShock exposure (t-1) x Leverage (t-1) | | | -0.0186 (0.164) |
| Leverage (t-1) | -0.0007 (0.0057) | | -0.0029 (0.0058) |
| Coherence (t-1) | 0.280*** (0.0072) | 0.291*** (0.0072) | 0.288*** (0.0074) |
| Year dummies | YES | YES | YES |
| Firm FE | YES | YES | YES |
| Observations | 22,969 | 22,815 | 21,939 |
| R-squared | 0.881 | 0.885 | 0.885 |

Mean (s.e.) coefficient estimates of panel regressions with firm fixed effects.

*$p < 0.1$

**$p < 0.5$

***$p < 0.01$

the process.). Checking the effect of the leverage ratio and its interaction with the ImpShock exposure variable, we observe that they are not significant and do not alter the positive effect of ImpShock exposure. Thus, the demand shock from import pressures is not moderated by the tightening of financial resources (Table 3 Column 3).

In sum, these results inform us that technological relatedness is a major factor of product portfolio diversification in firms that already have a specialized and technologically coherent product portfolio. These companies follow a strategy that exploits economies of scope. During a financial crisis, negative demand shocks foster specialization and portfolio coherence even further only for firms having financial liabilities. This suggests that companies tend to streamline their production in crisis times probably motivated by cost concerns.

The effect of exposure to market shrinkage or to import shock on the coherence of the firm's product portfolio may take more years to manifest. Therefore, we test if adding lags of 2 and 3 years of the shock variables change the overall picture of effects, i.e. if we see a dynamically changing effect over years (Table 4). In these regressions, we can only use those firms that have survived for at least four years, which means that our previous sample is cut to half. The fixed-effects regression on the coherence of the next year's product portfolio are consistent with the baseline specifications showing that exposure to market shrinkage is only significant with its interaction to the leverage ratio. However, Shrinkage exposure and ImpShock exposure show different dynamics in their effect on the product portfolio. In case of Shrinkage exposure, a considerable part of the effect is maintained for the 2nd year, whereas in case of ImpShock exposure, the effect rather falls back dramatically for the 2nd year. No significant effects can be seen in the 3rd year.

## 4.3. Narrowing and expanding the product portfolio

Regression results in which we investigate the events of termination of products from company portfolios are displayed in Table 5. We test two models, both of which include the Relatedness of the product to the core product of the company, the shock variables and their

**Table 4. Dynamics of the Coherence of firm's portfolios.**

| Dependent: | Coherence (t) | |
|---|---|---|
| Shrinkage exposure (t-1) | -0.0028 (0.0045) | |
| Shrinkage exposure (t-2) | -0.0015 (0.0048) | |
| Shrinkage exposure (t-3) | -0.0007 (0-0048) | |
| ImpShock exposure (t-1) | | 0.224*** (0.0506) |
| ImpShock exposure (t-2) | | -0.314*** (0.0492) |
| ImpShock exposure (t-3) | | -0.0035 (0.0507) |
| Shrinkage exposure (t-1) x Leverage (t-1) | 0.0341*** (0.0112) | |
| Shrinkage exposure (t-2) x Leverage (t-2) | 0.0255** (0.0117) | |
| Shrinkage exposure (t-3) x Leverage (t-3) | 0.0123 (0.0115) | |
| Leverage (t-1) | 0.0873 (0.0089) | |
| Leverage (t-2) | -0.0150* (0.0091) | |
| Leverage (t-3) | -0.0021 (0.0087) | |
| Coherence (t-1) | 0.127*** (0.0101) | 0.134*** (0.0101) |
| Year dummies | YES | YES |
| Firm FE | YES | YES |
| Observations | 13,042 | 12,899 |
| R-squared | 0.873 | 0.880 |

Mean (s.e.) coefficient estimates of panel regressions with firm fixed effects.

*p<0.1

**p<0.5

***p<0.01

**Table 5. Dropping products from firm's portfolios.**

| | Dependent variable: dropping a product (t) | |
|---|---|---|
| Relatedness (t-1) | -0.0846*** | -0.0913*** |
| | (0.0086) | (0.00992) |
| Market shrinkage | 0.0085 | |
| | (0.0052) | |
| Market shrinkage × Relatedness (t-1) | 0.0067 | |
| | (0.0172) | |
| Market shrinkage × Leverage × Relatedness (t-1) | 0.0141 (0.0392) | |
| Market shrinkage × Leverage | -0.0268** (0.0123) | |
| Leverage | -0.283 (0.0151) | |
| Import shock | | 0.627*** |
| | | (0.0659) |
| Import shock × Relatedness (t-1) | | -1.056*** |
| | | (0.216) |
| Year dummies | YES | YES |
| Firm FE | YES | YES |
| Observations | 77,018 | 64,950 |
| R-squared | 0.295 | 0.319 |

Mean (s.e.) coefficient estimates of panel regressions with firm fixed effects.

*p<0.1

**p<0.5

***p<0.01

interaction with the Relatedness indicator are introduced separately. In case of the shock associated with decreasing markets, the variables are also interacted with the Leverage ratio.

We find that the coefficient of relatedness of the product to the core product of the firm is significant in both cases. Its direction corresponds to the expectation that firms tend to drop those products with higher likelihood, which are less related to their core activity. A further finding suggests that firms are more likely to drop products, which were exposed to Import shock but there is no such effect of Market shrinkage. The interaction term of Relatedness and negative demand shock variables inform us whether firms are more likely to drop their products unrelated to their core product when experiencing negative shocks compared to other cases. This interaction term is insignificant in case of the Market shrinkage variable (Column 1), but it turns negative and significant in case of the Import shock variable. This latter result indicates that firms are more likely to drop an unrelated product in case Chinese import to EU intensifies in this given product. Considering the interaction of the Market shrinkage variable with the Leverage ratio, we see a negative coefficient, suggesting that more indebted firms are more likely to drop any declining demand products. The non-significant triple interaction with the relatedness variable however indicates that these products do not tend to be the more unrelated ones. That is, in times of financial crisis, companies drop products from their portfolios that are more exposed to the contraction of financial resources, but not unrelated products: they drop products for which customer demand is more likely to fall in response to the contraction of financial resources. However, related diversification of firms motivated by economies of scope are intensified by increasing competition from import. In sum, the import shock approach supports Hypothesis 2, but the market shrinkage approach does not.

Table 6 reports results of regressions, in which adding new products is the dependent variable. The estimation strategy is identical with the one from the previous table.

The positive and significant coefficient of Relatedness suggests that firms do not pick new products randomly but add those ones, which are related to their core business. This finding is

**Table 6. The effect of crisis on adding new products.**

|  | Dependent variable: adding a product (t+1) | |
| --- | --- | --- |
| Relatedness (t-1) | 0.152*** | 0.153*** |
|  | (0.0037) | (0.0005) |
| Shrinkage exposure' (t) | -0.00004 |  |
|  | (0.00017) |  |
| Shrinkage exposure' (t) × Relatedness (t-1) | 0.0356** |  |
|  | (0.0139) |  |
| Shrinkage exposure' (t) × Leverage (t) × Relatedness (t-1) | -0.0198 (0.0320) |  |
| Shrinkage exposure' (t) × Leverage (t) | -0.0005 (0.0005) |  |
| Leverage (t) | -0.00008 (0.00015) |  |
| ImpShock exposure' (t) |  | 0.0016* |
|  |  | (0.0009) |
| ImpShock exposure' (t) × Relatedness (t-1) |  | -0.354*** |
|  |  | (0.0134) |
| Core product FE | YES | YES |
| Time dummies | YES | YES |
| Observations | 6,918,673 | 5,934,154 |
| R-squared | 0.020 | 0.019 |

Mean (s.e.) coefficient estimates of panel regressions with fixed effects.

in line with previous studies on firm diversification [8, 9]. However, we find very small influence of the negative demand shock variables.

Falling demand in the markets of its existing products, that we measure by Shrinkage exposure, does not seem to deter firms from expanding their portfolios, while experiencing import shocks only does so to a minor extent. The interaction term between Shrinkage exposure and Relatedness is positive and significant and informs us that technological relatedness plays an important role here. In the case of falling demand in existing products, firms are likely to diversify into new products that are technologically related to their core product. This result supports Hypothesis 3, and appears to be independent of the indebtedness ratio of the firms, as all the interactions with Leverage ratio are insignificant.

The ImpShock exposure variable and its interaction term with Relatedness behaves differently. We find a positive and significant main effect of import shock and a negative significant interaction. Consequently, Hungarian firms tend to diversify to new unrelated products when their production must face an increasing import competition from China in the European markets. This result contradicts with the expectation formulated in Hypothesis 3 and indicates that firms follow a strategy of risk reduction that might help them to mitigate the increasing competition in their existing markets.

## 5. Conclusions

The literature of product diversification distinguishes between related diversification, when the firm expands into technologically similar product lines, and unrelated diversification, when these similarities do not exist. Related diversification is explained by efficiency arguments, whereas unrelated diversification is justified with risk mitigation, individual managerial motives and market-power-based drivers.

This paper provides new evidence on how technological relatedness influences firms' decisions on product portfolio when corporations' markets are hit by a demand shocks, thereby contributing to a better understanding both of firm level product diversification decisions and corporate behavior. Times of falling demand and increasing import competition provide great opportunities to observe the dynamics of firm diversification, because firms must adjust production by either narrowing or diversifying the product portfolio.

To analyze the potential impact of falling demand and increasing import competition on product diversification of firms we apply two methods. In the first exercise, we look at how coherence in terms of technological relatedness in the product portfolio changes in firms when they face market shrinkage or increasing import competition. In the second method, we analyze the relationship between the relatedness of the probability of dropping and adding a non-core product from the product portfolio in these scenarios.

Results of the first method indicate an increasing coherence of the portfolio after a firm was exposed to increasing import competition. However, when firms experience a shrinking market, those firms increase product portfolio coherence in response to falling demand, which also have significant financial exposure (high Leverage).

This provides useful lessons for understanding the behavior of firms in times of economic crisis. The financial crisis is also causing a demand shock, as is increasing import pressure. But in a financial crisis, there is another direct mechanism through which firms are affected: the lack of financial resources. So, in a financial crisis, there is a demand shock mechanism, captured by our Market shrinkage variable, and there is a liquidity shock mechanism, captured by our debt leverage variable, and we also look at the interaction of the two.

The results show that in the case of an import shock, there is no effect on the debt leverage indicator – even in interaction – because there is no liquidity shock mechanism. One could

expect that the demand shock mechanism alone would have a similar effect on corporate decisions in both cases and that the other liquidity mechanism in the financial crisis might amplify it. However, this is not exactly what we see, but that the demand shock mechanism does not have an effect in the case of a financial crisis, only for firms that have effective short-term liabilities. This difference, i.e. why the demand shock mechanism itself does not work in the same way in the case of a financial crisis as in the case of import shocks, may be explained by the different expectations of firms. During financial crisis households and firms postpone purchases in the short term due to a lack of financial resources causing only a short-term demand shock. Therefore, firms do not start to rearrange their product portfolios immediately unless the contraction of financial resources creates an additional incentive to cut costs. But in case of import shock, firms expect a long-term demand shock, as cheap Chinese import pressure is not expected to be short-term.

These findings suggest that falling demand and increasing competition limit the options and alters the motivations of managers by making the cost efficiency concerns of related diversification more important than risk-sharing concerns or individual managerial motives of unrelated firm expansion.

However, we observe different mechanisms behind increasing coherence in case of decreasing product markets and increasing import competition. This is consistent with the view that when exposed to decreasing product markets, the strengthening of the role of relatedness was supported by decisions on adding products to the portfolio. When product markets declined and firms choose to expand their portfolio, they took into account the efficiency gains of joint production of related products more, than in good times. In contrast, when exposed to strengthening import competition, results imply that coherence increases due to firms' behavior of dropping unrelated products. Furthermore, in the case of increasing import competition, we also observe a strong degree of unrelated diversification that is a signal of mitigating the risks by diversifying into something new.

These results are consistent with previous finding of the literature stressing that cost reducing strategies are the most frequent answer of firms to economic crisis [68, 69]. Kitching et al. [30] argue that some firms perceive crisis periods as opportunities to expand into new markets. However, they also emphasize that not every firm are able to implement such expanding strategy during crisis, because it requires resources. Firms hit hard by the crisis have limited resources, tend to focus rather on short–term survival, and are more likely to choose cost-cutting strategies. Indeed, we find that the decline of firm's product markets decreases the likelihood that the firm expands to new products but the expanding firms chose products that are related to their core production.

Our results are also in line with previous literature on firm behavior that argue for a tendency to combine retrenchment strategies (through rational cost- and asset-cutting and more efficient exploitation of existing resources) and investment strategies (through exploring new market opportunities and potential product innovations) [30]. We find that increasing import competition do not prevent firms adding new products, but it increases the rate of replacing them. In such situations, they drop products, but also add new ones. This is consistent with the findings of Bernard et al. [89] who show that such product switching depends on the revenue of firms and the nature of demand shocks. Firms with increasing revenue are likely to add and not likely to drop products; while positive demand shocks induce product expanse but negative shocks foster narrowing the portfolio.

## Author Contributions

**Conceptualization:** Károly Miklós Kiss, László Lőrincz, Balázs Lengyel.

**Data curation:** László Lőrincz, Zsolt Csáfordi.

**Formal analysis:** Károly Miklós Kiss, László Lőrincz, Zsolt Csáfordi, Balázs Lengyel.

**Funding acquisition:** Károly Miklós Kiss.

**Investigation:** Károly Miklós Kiss, László Lőrincz, Zsolt Csáfordi, Balázs Lengyel.

**Methodology:** Károly Miklós Kiss, László Lőrincz, Zsolt Csáfordi, Balázs Lengyel.

**Project administration:** Károly Miklós Kiss.

**Writing – original draft:** Károly Miklós Kiss, László Lőrincz, Zsolt Csáfordi, Balázs Lengyel.

**Writing – review & editing:** Károly Miklós Kiss, László Lőrincz, Balázs Lengyel.

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
