## [Decision Letter · Decision Letter 0]

11 Jan 2022

PONE-D-21-36675Related and unrelated firm diversification in crisis and in prosperityPLOS ONE

Dear Dr. Kiss,

Thank you for submitting your manuscript to PLOS ONE. After careful consideration, we feel that it has merit but does not fully meet PLOS ONE’s publication criteria as it currently stands. Therefore, we invite you to submit a revised version of the manuscript that addresses the points raised during the review process. In particular, the authors must address the concern of the first reviewer about the nature of the crisis and its effects on the product portfolios and the empirical strategy to pinpoint the effects of the crisis.I also recommend to clarify the limits of the relatedness index and to better discuss the external validity of the findings. Furthermore, as suggested by the second reviewer, the authors must better align the empirical analysis with the theoretical framework. In particular, additional robustness checks are needed since endogeneity is a concern. In general, make sure you address all the points raised by the referees in your rebuttal letter.

We look forward to receiving your revised manuscript.

Kind regards,

Massimo Riccaboni

Academic Editor

PLOS ONE

Journal Requirements:

Reviewers' comments:

Reviewer's Responses to Questions

**Comments to the Author**

1. Is the manuscript technically sound, and do the data support the conclusions?

Reviewer #1: Partly

Reviewer #2: Partly

2. Has the statistical analysis been performed appropriately and rigorously? 

Reviewer #1: No

Reviewer #2: N/A

3. Have the authors made all data underlying the findings in their manuscript fully available?

Reviewer #1: No

Reviewer #2: No

4. Is the manuscript presented in an intelligible fashion and written in standard English?

Reviewer #1: Yes

Reviewer #2: Yes

5. Review Comments to the Author

Reviewer #1: The aim of the paper is to investigate empirically how the product portfolios of firms change “in times of crisis and prosperity”. The authors find that product portfolios become more focused in the period 2005-2012, also by dropping so-called peripheral products. The empirical strategy is mainly based on panel fixed effects methods.

The research question is certainly relevant, but I have four main concerns on this paper’s framework:

A) The nature of the crisis and its effects on the product portfolios. The authors explicitly assume that the crisis in 2008-2009 affected firms through a demand shock, hence they empirically measure the intensity of the shock over time by looking at drops in sales (market shrinkage). Actually, the global crisis in 2008-2009 had its roots in shocks that propagated from financial markets, thus affecting the real economy through credit shortages for both firms and consumers. Thus, demand dropped as a response to a financial shock. In this context, the correlation between market shrinkage and product portfolios in the paper could be a spurious one. I suggest the author to control explicitly for heterogeneous financial exposure of firms and their impact on the changing composition of product portfolios. Market shrinkages can be the result rather than the driver of changing product portfolios during a crisis in 2008-2009. After testing explicitly for the specific nature of the crisis in 2008-2009, the authors could streamline better their hypotheses in the framework they propose.

B) Empirical strategy. The choice of the empirical strategy is most problematic. What the authors actually test is not the change of product portfolios by separating times of crisis from times of prosperity. (The ambiguity starts in the title, when they mention both “times of crisis and prosperity”.) As they implement a panel fixed effects with both firm and time dummies, the coefficients they find can only be interpreted as average effects along the entire period they analyze, 2005-2012, thus confounding both years of crisis and so-called prosperity. (Please note that markets can be volatile also in times of overall ‘prosperity’.) In this case, I suggest implementing an event study to check if there is actually an apparent impact on the combination of product portfolios in times of crisis that is different from regular times. In this case, see also the specific comment on the nature of the crisis from point (A), and its economic mechanism having an impact on firms’ decisions.

C) Technological relatedness. The authors mainly rely on a network measure proposed in an unpublished working paper by Neffke and Henning in a working paper series. The main problem of that measure is that it does not actually catch direct technology relationships. It assumes that if products are produced by the same firms, then they are likely related in productive technologies needed to produce them. Such a measure is a sort of ‘revealed technology relatedness’, which is possibly biased by the presence of: 1) conglomerated firms, which decide to differentiate their production lines in structurally unrelated markets; 2) vertically-integrated firms, which decide to produce the inputs they need for their output. The suggestion is to discuss these limits and possibly implement an augmentation of the binomial regression model of eq. 1, to take care of the above-mentioned biases. Right now, it is not clear to me what is the economic role of the covariates of eq. 1 (number of active firms, average profit level, income level, firm size in employees, and value added).

D) External validity. The framework the authors adopt implicitly considers any economic crisis as having the same impact on product portfolios. I think it would be useful to comment if such a framework could be valid when one confronts with economic crises that have a different origin. Can findings related to a financial crises be extended to other sorts of crises, like the latest pandemic crisis? Can the search for different (foreign) markets prevail, hence bringing more diversification in products, when the crisis is more national and less global in nature?

Minor comments:

- The paper in general reads well, but it lacks conciseness starting from the abstract. It is not immediately clear what the research question is. There are many statements or claims that are not actually investigated in the paper (e.g. “what conditions drive related versus unrelated diversification of firms, is still poorly understood.”). The conclusions even relate the work to pieces of literature that were not discussed in the literature review, e.g., trading firms and trade policy, although the paper does not specifically investigate exporters or importers. I suggest cutting unnecessary digressions.

- Some descriptive statistics of the sample could help. How many firms in each industry? How representative they are of the entire population? I understand that the same data have been often used in other published economic papers. Please provide references for data validation.

- Please provide a better resolution of graphs.

Reviewer #2: Past financial crises have triggered an interesting strand of literature about how financial and economic conditions shape firms’ decision and their product portfolio due among other factors to the re-allocation of resources, availability of slack resources and financial constraints.

This study has potential for making an interesting contribution to the existing strand of literature by studying the role of crises in firms’ diversification strategies. Using the Hungarian PRODCOM database, the paper advances that firms’ product portfolio became more technological related for firms that were exposed to demand shocks.

The main finding is in line with efficiency considerations but that it is not so surprising. I encourage the authors to revise the paper in the following possible directions:

Data and Methodology: the authors provide descriptive statistics about revealed relatedness, coherence, market shrinkage and Import shocks. It would be interesting to know more about the characteristics of firms and industries included in the sample. In particular:

i) The authors selected the period 2005-2012 whereas the PRODCOM database covers the period 2003-2012 so the authors may provide a brief explanation about the sample construction choices.

ii) It is interesting to better point out the number and type of booms and busts that you observe in the period of analysis as these market fluctuations may be attributed to different stimuli that are likely to affect industries/firms in a different way.

Empirically, the paper will also benefit from providing robustness to alternative operationalizations of economic crisis through measures that best capture shift in demand for example.

At page 24, the authors argue that Coherence of the previous year is still the strongest predictor of Coherence at time t, I was then wondering whether and how the economic crisis could really play a role, as so far does not seem to be a strong predictor and this may require a realignment between the theory and the empirics.

The fact that in Table 4 and 5 Market Shrinkage and Import Shock have opposite sign may also require additional explanation/interpretation.

At page 17, the authors highlight a potential problem of endogeneity which probably call for additional explanations.

About the Import measure, I was wonder whether you observe increasing import competition in general and not only during a crisis. If that is the case what you observe with this variable may not be related to a crisis event, this would probably require some additional explanations.

iii) It is also interesting to know more about how many firms are included in the sample and what are their characteristics for example in terms of size, R&D expenditures, age, diversification over the period, performance indicators. Size appears to be a very important characteristics as if your sample include mostly small companies, it is rather trivial that their best choice is go for efficiency solutions and opt for a coherent product portfolio since they are not deep pocket and suffer from liability of smallness. In this line, it is interesting to provide some measure about slack resources or financial constraints of these firms which may also be used as controls in the regressions.

iv) The authors also point out, at page. 7, that extant literature showed that in R&D intensive industries unrelated diversification is frequently applied therefore I wonder whether the authors experiment with the regression by using R&D intensity as a possible control.

v) At page 14, the authors make an example about the products in calculating the technological relatedness, what if the same product is in two industries? Are these cases possible or frequent? I would also include a real case example picked from your sample.

In general, when describing data and measure used it would be good to provide some examples. For instance, by picking up some companies including in your sample (eventually in an anonymized way) and provide some descriptive for the focal firm or informing the readers about the exact variables available at PRODCOM.

Theory: from a theoretical point of view I would include the discussion on opportunity cost looking for example at the work by (Aghion and Saint-Pauk, 1998; Berchicci et al., 2013; Steenkamp and Fang, 2011). (opportunity cost vs. financial cost strand of literature) in contrast with the financial constraint argument.

Conclusions: as the authors point out in page 9, “Despite cost reduction is the most frequent strategy of firms in economic crisis, especially in the short term; Some authors debate its effectiveness”; it would be interesting to explore whether firms that opt for unrelated diversification during burst had a higher payoff in terms for example of market value or other key performance indicators during booms. Provide additional information about the performances of these firms after a crisis, conditional on their diversification strategies, would strength the managerial and practical implications of the authors’ study, which are not included in the current version.

Thanks for the opportunity to read your paper.

6. PLOS authors have the option to publish the peer review history of their article (what does this mean?). If published, this will include your full peer review and any attached files.

Reviewer #1: No

Reviewer #2: No

---

## [Author Response · Author response to Decision Letter 0]

16 Aug 2022

Response to the Reviewers

Dear Editor and Reviewers,

We appreciate the opportunity to revise and resubmit our paper “Related and unrelated firm diversification in crisis and in prosperity”. We are thankful for the comments of the reviewers that enabled us to develop our paper more solid theoretically and empirically. We have revised the manuscript along two major lines. First, we reframed our theoretical approach that better corresponds to our empirical strategy, and second, we included an important factor to the analysis suggested by both reviewers, the liquidity constrains. We aimed to respond to each reviewer comments and implement them to the best of our knowledge. We are addressing each of the concerns and comments raised step by step in the followings.

Reviewer #1

Major concerns:

A) The nature of the crisis and its effects on the product portfolios. The authors explicitly assume that the crisis in 2008-2009 affected firms through a demand shock, hence they empirically measure the intensity of the shock over time by looking at drops in sales (market shrinkage). Actually, the global crisis in 2008-2009 had its roots in shocks that propagated from financial markets, thus affecting the real economy through credit shortages for both firms and consumers. Thus, demand dropped as a response to a financial shock. In this context, the correlation between market shrinkage and product portfolios in the paper could be a spurious one. I suggest the author to control explicitly for heterogeneous financial exposure of firms and their impact on the changing composition of product portfolios. Market shrinkages can be the result rather than the driver of changing product portfolios during a crisis in 2008-2009. After testing explicitly for the specific nature of the crisis in 2008-2009, the authors could streamline better their hypotheses in the framework they propose.

Thank you for drawing attention to this important issue. We included this argument to Section 2.3. regarding theories. This also made us modifying H1, breaking it down into H1a about the impact of negative demand shock in general, and H1b about the impact of negative demand shocks coupled with high financial exposure. Correspondingly, we included the financial exposure variable (together with its interaction with negative demand shocks) throughout the models in the paper. 

B) Empirical strategy. The choice of the empirical strategy is most problematic. What the authors actually test is not the change of product portfolios by separating times of crisis from times of prosperity. (The ambiguity starts in the title, when they mention both “times of crisis and prosperity”.) As they implement a panel fixed effects with both firm and time dummies, the coefficients they find can only be interpreted as average effects along the entire period they analyze, 2005-2012, thus confounding both years of crisis and so-called prosperity. (Please note that markets can be volatile also in times of overall ‘prosperity’.) In this case, I suggest implementing an event study to check if there is actually an apparent impact on the combination of product portfolios in times of crisis that is different from regular times. In this case, see also the specific comment on the nature of the crisis from point (A), and its economic mechanism having an impact on firms’ decisions.

Thank you for raising our attention to this issue. After reading your comments and re-reading the paper, the discrepancy between the introduction and the empirics also became apparent to us, that is based on the title, and the arguments in the introduction (and partly also in the theory), the reader would expected an empirical strategy that compares firm behavior over different time periods. 

To resolve this issue, we decided to fit the introduction and theory part to the empirics, and not the other way around. We did so, as we believe that a key contribution of our paper is the idea that market trends can be quite different across product markets, and those can be utilized for identification of firm behavior. In case we compared different time periods, we could not utilize this heterogeneity, and would loose both in terms of empirical validity and statistical power.

According to this, we re-framed the introduction, removing arguments about “times of crisis”, but emphasizing negative demand shocks that firms experience. In the theory section most arguments remained valid in this framing, but the ones about the strategies of firms in economic crisis we reconsidered. We also modified the title of the paper. 

C) Technological relatedness. The authors mainly rely on a network measure proposed in an unpublished working paper by Neffke and Henning in a working paper series. The main problem of that measure is that it does not actually catch direct technology relationships. It assumes that if products are produced by the same firms, then they are likely related in productive technologies needed to produce them. Such a measure is a sort of ‘revealed technology relatedness’, which is possibly biased by the presence of: 1) conglomerated firms, which decide to differentiate their production lines in structurally unrelated markets; 2) vertically-integrated firms, which decide to produce the inputs they need for their output. The suggestion is to discuss these limits and possibly implement an augmentation of the binomial regression model of eq. 1, to take care of the above-mentioned biases. Right now, it is not clear to me what is the economic role of the covariates of eq. 1 (number of active firms, average profit level, income level, firm size in employees, and value added).

This method does indeed measure a revealed technology relatedness, which is however a fairly common and accepted method in the literature if we cannot directly measure technology relationships. The description of the Neffke and Henning method has been published in several later applications in peer-reviewed journals (see e.g. Neffke F, Henning M., Boschma R. (2011): The impact of ageing and technological relatedness on agglomeration externalities: a survival analysis. Journal of Economic Geography, 12(2): 485-517.; Neffke F., Henning M., Boschma R. (2011): How do regions diversify over time? Industry relatedness and the development of new growth paths in regions. Economic Geography 87(3):237-265.; Neffke F., Henning M., Boschma R., Lundquist K.J. & Olander L.O. (2011). The dynamics of agglomeration externalities along the life cycle of industries. Regional Studies 45(1): 49-65.), but the most detailed description can be found in the referenced working paper, so it is usually referred to.

Presence of conglomerated and vertically-integrated firms would not be a problem if we had site-level data, as firms produce technologically related products at the same site, which can benefit from the resulting cost efficiency in production. Unfortunately, our database only contains firm-level data. However, two things reduce the problem raised. First, it is precisely to avoid this multi-plant problem that we are looking at manufacturing industries. Békés and Harasztosi (Békés, G. and Harasztosi, P, 2013. "Agglomeration premium and trading activity of firms," Regional Science and Urban Economics, Elsevier, vol. 43(1), 51-64.) show that in the Hungarian manufacturing industries over 90% of firms have one site only, which suggests that the bias is not significant. Moreover, as shown in the description of the method, the difference between the predicted and observed co-occurrences reveals the relatedness between each industry pair. The relatedness indicator between two industries would be significantly distorted if both industries were composed of very similar conglomerates or vertically integrated companies expanding in the same unrelated directions. Thus, the problem is considered legitimate, but the arguments above suggest that the bias does not really give cause for concern. Nevertheless, we thank for drawing our attention to this necessary addition, and we have included a discussion of this bias in the description of the method.

Regarding the economic role of the covariates in equation 1, as we discussed above the equation, diversification into products can be motivated not only by economies of scope and technological proximity. Other factors, such as high expected profits or industry size are important as well so that firms might decide to expand their activities to attractive industries. To control for other factors that influence the number of co-occurrence links we followed Neffke and Henning (2008) and estimate the expected number of co-occurrences for each pair of industries using industry specific characteristics as predictors (e.g. profitability, sales volume, competition expectations, wage level etc.).

D) External validity. The framework the authors adopt implicitly considers any economic crisis as having the same impact on product portfolios. I think it would be useful to comment if such a framework could be valid when one confronts with economic crises that have a different origin. Can findings related to a financial crises be extended to other sorts of crises, like the latest pandemic crisis? Can the search for different (foreign) markets prevail, hence bringing more diversification in products, when the crisis is more national and less global in nature?

We examine the impact of demand shocks in general, which can arise for several reasons. In the period under study, there were several such causes, e.g. EU accession, import shocks and financial crisis, all of which have in common that they cause a demand shock. However, in case of financial crisis there is an additional mechanism, which we have examined separately in the new version. We tried to clarify this in the paper. And yes, all of this makes us think that our results can apply to any crisis that causes a demand shock.

Minor comments:

- The paper in general reads well, but it lacks conciseness starting from the abstract. It is not immediately clear what the research question is. There are many statements or claims that are not actually investigated in the paper (e.g. “what conditions drive related versus unrelated diversification of firms, is still poorly understood.”). The conclusions even relate the work to pieces of literature that were not discussed in the literature review, e.g., trading firms and trade policy, although the paper does not specifically investigate exporters or importers. I suggest cutting unnecessary digressions.

Thank you for your suggestion. We have tried to clarify the research questions and focus our message better.

- Some descriptive statistics of the sample could help. How many firms in each industry? How representative they are of the entire population? I understand that the same data have been often used in other published economic papers. Please provide references for data validation.

We inserted data to the text in section 3.1, and also a new figure (Figure 2) on the distribution of firms by size and industry, and comparison to official publication (Eurostat). 

- Please provide a better resolution of graphs.

Reviewer #2

i) The authors selected the period 2005-2012 whereas the PRODCOM database covers the period 2003-2012 so the authors may provide a brief explanation about the sample construction choices.

Thank you for pointing this out. In fact we utilize the data for the analysis from the whole period of 2003-2012; so we modified the description accordingly. What might have lead to this mistake in the description is that we use both a lagged and forward term in the estimation, which decreases the number of effective years used for identifications by two. 

ii) It is interesting to better point out the number and type of booms and busts that you observe in the period of analysis as these market fluctuations may be attributed to different stimuli that are likely to affect industries/firms in a different way.

Thank you, we added a related description to the Introduction (at line 100).

Empirically, the paper will also benefit from providing robustness to alternative operationalizations of economic crisis through measures that best capture shift in demand for example.

At page 24, the authors argue that Coherence of the previous year is still the strongest predictor of Coherence at time t, I was then wondering whether and how the economic crisis could really play a role, as so far does not seem to be a strong predictor and this may require a realignment between the theory and the empirics.

Thank you for this comment. We modified the description of the results here, removing the text that previous year was the strongest predictor – which is in fact true statistically, however this does not have theoretical relevance, as the coherence of previous year is only a technical variable to serve as reference point for the next year and to capture the stability of the process.

The fact that in Table 4 and 5 Market Shrinkage and Import Shock have opposite sign may also require additional explanation/interpretation.

At the suggestion of the other reviewer, we included a financial exposure variable (together with its interaction with negative demand shocks) throughout the models. This changed the results slightly, but there was still a difference between the effects of the Market Shrinkage and Import Shock variables. This difference is explained in more detail on page 33 and also in Conclusions on pages 37-38.

At page 17, the authors highlight a potential problem of endogeneity which probably call for additional explanations.

We have supplemented the explanation.

About the Import measure, I was wonder whether you observe increasing import competition in general and not only during a crisis. If that is the case what you observe with this variable may not be related to a crisis event, this would probably require some additional explanations.

Import pressures are another possible source of demand shocks in product markets, measured over the entire time window under study. We have tried to clarify this at several points in the paper.

iii) It is also interesting to know more about how many firms are included in the sample and what are their characteristics for example in terms of size, R&D expenditures, age, diversification over the period, performance indicators. 

In the new version we included the information on distribution of the firms included in terms of size and industry to section 3.1 (and Figure 2). Unfortunately, our data do not cover R&D expenditures and the age of firms, so we could not include these. We also supplemented the data description with information on size and performance indicators (Table 1 in section 3.1)

Size appears to be a very important characteristics as if your sample include mostly small companies, it is rather trivial that their best choice is go for efficiency solutions and opt for a coherent product portfolio since they are not deep pocket and suffer from liability of smallness. In this line, it is interesting to provide some measure about slack resources or financial constraints of these firms which may also be used as controls in the regressions.

Thank you for this suggestion. In line with this suggestion, and with the related one of Reviewer 1 to address financial constraints directly, we added the analysis of financial constraints to the analysis throughout the article, and also modified H1, breaking it down into two sub-hypotheses; H1a about the impact of negative demand shock in general, and H1b about the impact of negative demand shocks coupled with high financial exposure.

iv) The authors also point out, at page. 7, that extant literature showed that in R&D intensive industries unrelated diversification is frequently applied therefore I wonder whether the authors experiment with the regression by using R&D intensity as a possible control.

Thank you for this suggestion. However, we could not implement this, as we do not have data about R&D expenditures. (R&D data would only be available from the European Innovation Survey, but that considers a more limited sample that only partly overlaps with the Prodcom.) 

v) At page 14, the authors make an example about the products in calculating the technological relatedness, what if the same product is in two industries? Are these cases possible or frequent? 

In our approach it is not possible for a product to belong to two different industries, as we assign each firm the industry, to which its core product belongs to (when the industry is defined by the 4 digits of the 6-digit product code). As the product classification is hierarchical, it is not possible that the same product code appears under two separate industry codes. We describe this procedure in paragraphs 2-4 of section 2.3. Note that with this method we in fact disregard the registered industry code of the firms, to which they classified themselves; and “correct” that based on their actual production behaviour. 

I would also include a real case example picked from your sample.

In general, when describing data and measure used it would be good to provide some examples. For instance, by picking up some companies including in your sample (eventually in an anonymized way) and provide some descriptive for the focal firm or informing the readers about the exact variables available at PRODCOM.

Thank you, we agree that this would indeed be informative. However, we are unable to do that due to the explicit restriction in our contract with the Central Statistical Office that forecloses reporting any information that refers to the activity of a single individual unit (firm), even in an anonymised way. 

Theory: from a theoretical point of view I would include the discussion on opportunity cost looking for example at the work by (Aghion and Saint-Pauk, 1998; Berchicci et al., 2013; Steenkamp and Fang, 2011). (opportunity cost vs. financial cost strand of literature) in contrast with the financial constraint argument.

The proposed line of literature gives the same cost-effectiveness argument, starting from the same literary roots (Penrose, Teece) as we use, but using slightly different language.

Conclusions: as the authors point out in page 9, “Despite cost reduction is the most frequent strategy of firms in economic crisis, especially in the short term; Some authors debate its effectiveness”; it would be interesting to explore whether firms that opt for unrelated diversification during burst had a higher payoff in terms for example of market value or other key performance indicators during booms. Provide additional information about the performances of these firms after a crisis, conditional on their diversification strategies, would strength the managerial and practical implications of the authors’ study, which are not included in the current version.

These are interesting questions but are beyond the scope of this paper and would require a separate study.

---

## [Decision Letter · Decision Letter 1]

25 Oct 2022

PONE-D-21-36675R1Related adjustment of firm production in economic crisisPLOS ONE

Dear Dr. Kiss,

Thank you for submitting your manuscript to PLOS ONE. After careful consideration, we feel that it has merit but does not fully meet PLOS ONE’s publication criteria as it currently stands. Therefore, we invite you to submit a revised version of the manuscript that addresses the points raised during the review process.

We look forward to receiving your revised manuscript.

Kind regards,

Massimo Riccaboni

Academic Editor

PLOS ONE

Journal Requirements:

Reviewers' comments:

Reviewer's Responses to Questions

**Comments to the Author**

1. If the authors have adequately addressed your comments raised in a previous round of review and you feel that this manuscript is now acceptable for publication, you may indicate that here to bypass the “Comments to the Author” section, enter your conflict of interest statement in the “Confidential to Editor” section, and submit your "Accept" recommendation.

Reviewer #1: All comments have been addressed

Reviewer #2: (No Response)

2. Is the manuscript technically sound, and do the data support the conclusions?

Reviewer #1: Yes

Reviewer #2: Partly

3. Has the statistical analysis been performed appropriately and rigorously? 

Reviewer #1: Yes

Reviewer #2: (No Response)

4. Have the authors made all data underlying the findings in their manuscript fully available?

Reviewer #1: No

Reviewer #2: No

5. Is the manuscript presented in an intelligible fashion and written in standard English?

Reviewer #1: Yes

Reviewer #2: Yes

6. Review Comments to the Author

Almost all comments raised by the reviewers in the previous round have been address but there are a few remaining issues as listed below. Make sure you address them all in the next round of revision. 

Reviewer #1: The authors have properly addressed almost all issues raised in the previous referee reports. I would add just one concern and a suggestion.

My concern is that the authors should be more cautious in interpreting findings. They too often interpret them in terms of 'impact', and they explicitly state that they are sure to rule out endogeneity by just adding lagged values of regressors. I understand that this point was raised by the other referee. From an econometric perspective, lagged values in a panel data setting are just a naive solution to endogeneity, perhaps just a first step to address simultaneity (which is one possible story of endogeneity). Yet, the underlying economic relationships that the authors are testing are complex, and I would not rule out possible 'reverse causality' issues or a few omitted variable bias. The triad market structure/financial constraints/demand shocks has more than one possible direction of causality, as scholars of industrial organization or corporate finance could well explain. Eventually, my suggestion is to stay as much as possible with an interpretation in terms of correlations.

Reviewer #2: I have read the paper and I am happy with some of the changes the authors have implemented. I feel that the paper is going towards a more convincing path. Nevertheless, I do not find all the concerns were addressed satisfactorily and a few minor issues have emerged in the new version. I have few additional comments and suggestions.

- Terminology: I appreciate that the authors have revised some terminology in the manuscript using demand shocks, however there is lack of consistency to this terminology along the full paper which requires a more in depth rephrase of the sentences and reconceptualization along the entire paper (in some passages there is still reference to differences between crisis and prosperous times, or to the use of different synonyms.) I would suggest the authors to define exactly what they mean with economic crisis and stick to their definition of crisis/demand shock along the entire paper (they are going in that direction but needs some more editing).

- Type of burst: the type of burst or different types of crises/demand shocks raised in the previous round was not addressed satisfactory. I feel the authors have just added one sentences but the differences could be better described.

- Financial distress: I appreciated that the authors have added a variable that may capture the financial distress of the firms in their sample, and for consistency reasons I would like to see the full model in table 3. In the current version the authors argue that results are not altered therefore for consistency would be nice to report the full model.

- Additional statistics: in the previous round I suggested to include some statistics about the firms in the sample. I appreciate the efforts of the authors in addressing this suggestion, however it is not fully clear to me whether the statistics come from Eurostat or from their sample. They argue that the PRODCOM database includes approximately 7000 firms but in the following paragraph of Section 3.1. they indicate that there were 550 firms operating in the observed period based on Eurostat. I would better clarify this point, where the statistics come from and so forth. Moreover, although the authors do not have firm age, they could probably retrieve it from the date of incorporation, if available.

- Theory: I also appreciate the efforts in adjusting the theory. However, I think that the introduction and theory part could be further strengthen by problematize more the topic of research. This is also linked to the fact that the results that firms opt for related products, especially those that are financially distressed, is not surprising per se. Therefore, I would suggest to address more in depth the theory and the empirics in order to create a sort of tension or to problematize better this interesting issue. I also noticed that in some of the theory, the authors make express reference to how the theory would adapt in case of small firms. Extending the discussion to how the mechanisms derived from the theory cited in the manuscript would adapt in the context of small firms make sense if the sample is mostly based on small. If this is the case, I would suggest to adapt Authors prediction based also on how the mechanisms and theory they describe would change, the authors partially do so for example at the end of Section 2.2.

- Editing: I noticed that some of the new passages are a bit fragmented and there are some typos. Along this line, I noticed that in the abstract, intro and theory, the authors refer to the coherence measure using the term cohesiveness. As suggested in the first comment, also in this case, I would stick to the term coherence in order to make it consistent with the empirics. In table 3 I would clarify whether the variables are at t-1, I think so based on your equation N. 9, if so I would suggest adapt the table.

7. PLOS authors have the option to publish the peer review history of their article (what does this mean?). If published, this will include your full peer review and any attached files.

Reviewer #1: No

Reviewer #2: No

---

## [Author Response · Author response to Decision Letter 1]

15 Dec 2022

Response to the Reviewers

Dear Editor and Reviewers,

We appreciate the opportunity to revise and resubmit our paper “Related and unrelated firm diversification in crisis and in prosperity”. We are thankful for the comments of the reviewers that enabled us to develop our paper more solid theoretically and empirically. We are addressing each of the concerns and comments raised step by step in the followings.

Reviewer #1: The authors have properly addressed almost all issues raised in the previous referee reports. I would add just one concern and a suggestion.

My concern is that the authors should be more cautious in interpreting findings. They too often interpret them in terms of 'impact', and they explicitly state that they are sure to rule out endogeneity by just adding lagged values of regressors. I understand that this point was raised by the other referee. From an econometric perspective, lagged values in a panel data setting are just a naive solution to endogeneity, perhaps just a first step to address simultaneity (which is one possible story of endogeneity). Yet, the underlying economic relationships that the authors are testing are complex, and I would not rule out possible 'reverse causality' issues or a few omitted variable bias. The triad market structure/financial constraints/demand shocks has more than one possible direction of causality, as scholars of industrial organization or corporate finance could well explain. Eventually, my suggestion is to stay as much as possible with an interpretation in terms of correlations.

Thank you. Accepting the reviewer's suggestion, we have restated our conclusions.

Reviewer #2: I have read the paper and I am happy with some of the changes the authors have implemented. I feel that the paper is going towards a more convincing path. Nevertheless, I do not find all the concerns were addressed satisfactorily and a few minor issues have emerged in the new version. I have few additional comments and suggestions.

- Terminology: I appreciate that the authors have revised some terminology in the manuscript using demand shocks, however there is lack of consistency to this terminology along the full paper which requires a more in depth rephrase of the sentences and reconceptualization along the entire paper (in some passages there is still reference to differences between crisis and prosperous times, or to the use of different synonyms.) I would suggest the authors to define exactly what they mean with economic crisis and stick to their definition of crisis/demand shock along the entire paper (they are going in that direction but needs some more editing).

Thank you. Reading the manuscript in this light, it became apparent that in the previous version the arguments about crises and demand shocks were mixed. So we cleared this up by concentrating on demand shocks, and mentioning the crises only as a specific case of negative shocks in the corresponding section of the literature (2.3.)

- Type of burst: the type of burst or different types of crises/demand shocks raised in the previous round was not addressed satisfactory. I feel the authors have just added one sentences but the differences could be better described.

In the previous round we were struggling finding proper solution. As we identify from the heterogeneity over thousands of products, they have turned out to be way too many to create meaningful visualizations, and picking random examples just felt arbitrary. Still, it is absolutely a valid claim to give the reader the feeling about the heterogeneity and the related economic trends here. Accordingly, we added the following paragraph.

“The investigated period covers interesting heterogeneity of demand shocks across product markets and firms. Trade liberalization opened the Hungarian industrial market in the 1990s and has concluded after the country joined the EU in 2004. This opening hit some of the traditional industries. For example, a significant decline was observed in the food industry due to the adverse effect of EU agricultural agreements, and in the and textile industry due to a reallocation mechanism that destroyed less productive firms (Békés-Halpern-Muraközy 2011, Harasztosi 2011) and to a significant increase of the minimal wage (Harasztosi and Lindner 2019). The period after EU accession is associated with increasing foreign direct investment, gross value added, and export orientation, which is interrupted in 2008-2010 by the global financial crisis that also hit industries to a different extent. After the financial crisis, a re-industrialization can be observed that is selective across industries. Transport equipment sector grew both in terms of employment and value-added, however, computers and electronics have decreased (Lengyel 2011, Lengyel et al 2017).” 

- Financial distress: I appreciated that the authors have added a variable that may capture the financial distress of the firms in their sample, and for consistency reasons I would like to see the full model in table 3. In the current version the authors argue that results are not altered therefore for consistency would be nice to report the full model.

Thank you for the comment. We added that as Column 3 of Table 3.

- Additional statistics: in the previous round I suggested to include some statistics about the firms in the sample. I appreciate the efforts of the authors in addressing this suggestion, however it is not fully clear to me whether the statistics come from Eurostat or from their sample. They argue that the PRODCOM database includes approximately 7000 firms but in the following paragraph of Section 3.1. they indicate that there were 550 firms operating in the observed period based on Eurostat. I would better clarify this point, where the statistics come from and so forth. 

Thank you. We reorganized the corresponding two paragraphs and added more details to provide the readers a clearer overview on the process and the sources of data. 

Moreover, although the authors do not have firm age, they could probably retrieve it from the date of incorporation, if available.

Thank you. We finally obtained the date of incorporation from another data source. (Balance sheet data from the national tax authority) that we could merge to the balance sheets that we had from the statistical office, so we could include firm age to Table 1.

- Theory: I also appreciate the efforts in adjusting the theory. However, I think that the introduction and theory part could be further strengthen by problematize more the topic of research. This is also linked to the fact that the results that firms opt for related products, especially those that are financially distressed, is not surprising per se. Therefore, I would suggest to address more in depth the theory and the empirics in order to create a sort of tension or to problematize better this interesting issue.

Thank you. Now we have re-written the introduction to focus better on our contribution to the literature. We have also streamlined the theory, along your previous comment on the terminology, as we found that it was not only a terminology issue. We also added two papers, you recommended in the previous revisions (Aghion & Saint-Paul; Steenkapm & Fang) to improve the theory. 

I also noticed that in some of the theory, the authors make express reference to how the theory would adapt in case of small firms. Extending the discussion to how the mechanisms derived from the theory cited in the manuscript would adapt in the context of small firms make sense if the sample is mostly based on small. If this is the case, I would suggest to adapt Authors prediction based also on how the mechanisms and theory they describe would change, the authors partially do so for example at the end of Section 2.2.

Thank you. In fact, this is true for our sample, the majority of the firms are small, falling into the 10-50 employee category. This is however just the consequence of the distribution of firms, which we did not want to restrict further, beyond that is applied in the PRODCOM generally.

In this version we added a sentence along the specificities of small firms to Section 3.3. too, in addition what was already present in section 2.2. However, we did not want to elaborate this line of argument much further, because that would eventually lead to developing specific hypotheses for small versus large firms, where we would not want to drive this paper for both theoretical and practical reasons. On the theoretical side we observe that those papers referring to small firms have only small firms in their samples, and do not make comparisons by large ones, therefore we cannot easily develop consistent hypotheses to differentiate them. On the practical side we think that the current empirical examination having two shock mechanisms, the interaction of financial exposure, and the two adjustment channels of dropping and adding products is rich enough, so it would have been too much if we added a further interaction of firm size.

- Editing: I noticed that some of the new passages are a bit fragmented and there are some typos. Along this line, I noticed that in the abstract, intro and theory, the authors refer to the coherence measure using the term cohesiveness. As suggested in the first comment, also in this case, I would stick to the term coherence in order to make it consistent with the empirics. In table 3 I would clarify whether the variables are at t-1, I think so based on your equation N. 9, if so I would suggest adapt the table.

Thank you, we reviewed the text, got rid of the cohesiveness term and edited the tables to be consistent with the equations.

---

## [Editor Report · Decision Letter 2]

2 Jan 2023

Related adjustment of firm production after demand shocks

PONE-D-21-36675R2

Dear Dr. Kiss,

We’re pleased to inform you that your manuscript has been judged scientifically suitable for publication and will be formally accepted for publication once it meets all outstanding technical requirements.

Kind regards,

Massimo Riccaboni

Academic Editor

PLOS ONE

Additional Editor Comments (optional):

There are some typos left. Make sure a native speaker proofreads the paper before publication.

---

## [Editor Report · Acceptance letter]

6 Jan 2023

PONE-D-21-36675R2 

Related adjustment of firm production after demand shocks 

Dear Dr. Kiss:

I'm pleased to inform you that your manuscript has been deemed suitable for publication in PLOS ONE. Congratulations! Your manuscript is now with our production department. 

Kind regards, 

on behalf of

Dr. Massimo Riccaboni 

Academic Editor

PLOS ONE